# Epidermal Electrodes with Ferrimagnetic/Conductive Properties for Biopotential Recordings

**DOI:** 10.3390/bioengineering9050205

**Published:** 2022-05-11

**Authors:** Andrea Spanu, Mohamad Taki, Giulia Baldazzi, Antonello Mascia, Piero Cosseddu, Danilo Pani, Annalisa Bonfiglio

**Affiliations:** 1Department of Electrical and Electronics Engineering, University of Cagliari, Piazza D’Armi, 09123 Cagliari, Italy; mohamad.taki@unica.it (M.T.); giulia.baldazzi@unica.it (G.B.); antonello.mascia@unica.it (A.M.); piero.cosseddu@unica.it (P.C.); danilo.pani@unica.it (D.P.); annalisa.bonfiglio@unica.it (A.B.); 2Department of Electrical & Electronics Engineering, Lebanese International University, Beirut 146404, Lebanon; 3Department of Bioengineering, Robotics and System Engineering, University of Genoa, Via All’Opera Pia 13, 16145 Genova, Italy; 4Interdepartmental Center for Amyotrophic Lateral Sclerosis and Motor Neuron Diseases, 09100 Cagliari, Italy; 5Department of Science, Technology and Society, Scuola Universitaria Superiore IUSS Pavia, Palazzo del Broletto, Piazza della Vittoria 15, 27100 Pavia, Italy

**Keywords:** epidermal electrodes, ferrimagnetic conductive films, ECG recordings, PEDOT:PSS free-standing films, magnetic connectors

## Abstract

Interfacing ultrathin functional films for epidermal applications with external recording instruments or readout electronics still represents one of the biggest challenges in the field of tattoo electronics. With the aim of providing a convenient solution to this ever-present limitation, in this work we propose an innovative free-standing electrode made of a composite thin film based on the combination of the conductive polymer PEDOT:PSS and ferrimagnetic powder. The proposed epidermal electrode can be directly transferred onto the skin and is structured in two parts, namely a conformal conductive part with a thickness of 3 μm and a ferrimagnetic-conductive part that can be conveniently connected using magnetic connections. The films were characterized for ECG recordings, revealing a performance comparable to that of commercial pre-gelled electrodes in terms of cross-spectral coherence, signal-to-noise ratio, and baseline wandering. These new, conductive, magnetically interfaceable, and free-standing conformal films introduce a novel concept in the domain of tattoo electronics and can set the basis for the development of a future family of epidermal devices and electrodes.

## 1. Introduction

More than ten years after their introduction, epidermal (or tattoo) electronics have become a very interesting, multifaceted, and prolific area of research which embraces a wide variety of biomedical applications, ranging from biopotential monitoring to artificial electronic skin and in vivo cellular applications [1,2,3,4]. In particular, thanks to the clear advantages offered by their ultrathin, conformable nature, several epidermal sensors have been recently proposed to measure biosignals and physiological parameters, such as electrocardiography (ECG), electromyography, skin hydration, and human perspiration [5,6,7,8]. For most of these flexible electrodes and sensors, the interconnection with external readout systems for data acquisition and processing is a well-known problem [9], usually tackled by rigid and bulky connectors, which strongly limits the benefits from the use of imperceptible ultrathin devices. Several techniques have been proposed in the past decade to overcome this issue [10,11], some of which involve the use of magnetic connections. An interesting example of this approach is represented by the recent work of Jang et al. [12], who proposed a micro-structured conductive and elastic metallic mesh, which is folded around a magnetic rubber patch that can be connected using magnetic connectors. A different example is represented by the recent work of Greco et al. [13], who have proposed free-standing PEDOT:PSS-based conductive films connected with a magnet through a stretchable connection made of a silver ink–based glue. Although the aforementioned solutions present some advantages over standard approaches where epidermal electrodes are connected to the hard electronic components through, for example, wiring pads or clips [14,15], they also suffer from several disadvantages. In particular, the solution proposed in [11] gives rise to a quite thick ferrimagnetic folded electrode, and it cannot be defined as conformal as the use of the ferrimagnetic core leads to a total thickness on the order of 500 μm. As a consequence, an adhesive layer must be used in order to ensure the adhesion between the electrodes and the skin. On the contrary, the film proposed in [13], although ultra-conformable and free-standing, cannot be directly connected using magnetic connectors. As a matter of fact, the film has to be connected to an external electronic device using a silver ink–based acrylic glue, and the magnets present in the external device (which are glued on the opposite side of it) serve the sole purpose of ensuring the contact with the magnetic connector (which is placed on the top to ensure the electrical contact). Another different approach involving a magnetic connection is reported by Dai et al. [16], who proposed a very interesting functionalization and printing process to obtain flexible magneto–electrical films. The proposed films are electrically conductive and exhibit good magnetic properties but cannot be defined as conformable and must be integrated with actual epidermal patches for their use as biomedical devices for the recording of biopotentials, thus acting more like an interface between the sensor itself and the communication circuit. Moreover, in order to achieve the desired magnetic properties, a high-voltage step must be included in the fabrication process, thus making it not easily upscalable.

With the intent of overcoming the aforementioned issues and thus providing an innovative approach for an epidermal electrode interface, we developed a simple fabrication process with which it is possible to obtain ultrathin functional films that can be easily contacted using magnetic connectors. In particular, in this paper we present an alternative and easy way to fabricate free-standing conductive and ferrimagnetic films based on a blend of the conductive polymer PEDOT:PSS and PVA and inexpensive ferrite powder as the ferrimagnetic material. In the biomedical field, PEDOT:PSS is, in fact, one of the most interesting materials thanks to its electronic and ionic conductivity and the possibility of depositing it using low-cost techniques such as spin coating [17], inkjet printing [18], and screen printing [19]. Recently, this remarkable material was used for cellular interfaces (both in vivo and in vitro) [20], textile applications [21,22,23], and epidermal electronics [24]. The choice of blending conductive ink with PVA is twofold: first of all, thanks to its film-forming and emulsifying properties [25], the presence of PVA allows a more uniform deposition by spin coating; secondly, the integration of PVA in the film allows easy removal of the final assembly from the carrier substrate without the use of a sacrificial layer. At the same time, ferrite is extensively used for biomedical applications such as drug delivery and magnetic resonance imaging (MRI), mainly because of its ferrimagnetic properties, its inexpensiveness, and its biocompatibility [26,27,28]. Moreover, the simple approach presented in this work allows obtaining high-performing films without any chemical modification of the particles (such as that reported by Najafi et al. [29]). The result of the combination of PEDOT:PSS, PVA, and ferrite shown in this work is a free-standing thin film that can be transferred onto the skin as a temporary tattoo, composed of a conductive, ultrathin part and a thicker part, both conductive and ferrimagnetic, which can be directly connected using magnetic connectors. This solution allows preventing any film damage due to unwanted mechanical mismatches between the film and the connector, thus greatly simplifying both the measurement apparatus and the overall experimental procedure. It is also noteworthy that the presented fabrication method also allows an easy modulation of the intensity of the magnetic attraction depending on the content of ferrimagnetic powder in the final structure. Moreover, the advantages offered by this approach can be relevant not only in standard epidermal applications but also in emerging scientific fields such as the in vitro monitoring of 3D cellular assemblies [30,31,32], where a seamless connection with ultrathin devices is of the greatest importance.

## 2. Materials and Methods

### 2.1. Composite Film Materials

The composite films are made of a conductive PEDOT:PSS ink mixed with a PVA solution and ferrite powder and are fabricated using a layer-by-layer process which alternates the PVA/PEDOT:PSS ink blend and the ferrite powder. The conductive ink is prepared by mixing 70% in volume of PEDOT:PSS (Clevios PH1000, by Heraeus, Hanau, Germany) with 30% in volume of ethylene glycol (by Merck KGaA, Darmstadt, Germany), followed by the addition of 1% (in volume of the previous blend) of 3-Glycidyloxypropyl trimethoxy-silane—GOPS (by Merck KGaA, Darmstadt, Germany), which is used as cross-linker. The PVA solution is separately prepared by diluting the PVA (molecular weight: 205 kg/mol) in de-ionized water (10% by weight). One of the main drawbacks of using PVA in this context is its well-known detrimental effect on the electrical properties of PEDOT:PSS [33]. In order to maintain the advantages offered by the PVA without significantly compromising the electrical properties of the deposited layers, two different blends with different PVA solution percentages, namely 7% and 2% in volume, were employed in different steps of the fabrication procedure, as explained in the fabrication section.

### 2.2. Fabrication Process

In the first step of the fabrication process, the PEDOT:PSS/PVA blend with 7% PVA is deposited through spin-coating (spin speed = 700 rpm, spin time = 60 s) on a PET substrate, which acts as a carrier (Figure 1A). Such a high PVA concentration has been used only at this stage with the only purpose of allowing an easy removal of the final film from the PET carrier. After the deposition of the first layer, the samples are baked at 75 °C for 20 min. The ferrite powder is then deposited over one half of the film in order to obtain a two-part electrode: a thin conductive part for an optimal skin interface, and a thicker portion both conductive and ferrimagnetic, which can be used for the magnetic connection. To obtain a good distribution of the ferrite powder and minimize the formation of big particle clusters (which can make the eventual peeling off of the final film very complicated and poorly reproducible), the powder is deposited using a filter constituted by a thin perforated Parylene C film (thickness: 500 nm; hole diameter: 50 μm), as shown in Figure 1B.

After the ferrite deposition, the PEDOT:PSS/PVA mix with the lowest PVA concentration is spin-coated on the whole substrate (spin speed = 700 rpm, spin time = 60 s) to form a second layer, and then baked at 75 °C for 5 min. A lower PVA concentration allows retaining the PVA film-forming properties without significantly affecting the conductivity of the blend. The second and third steps of the process, i.e., ferrite deposition and subsequent PEDOT:PSS/PVA ink spin coating, can be repeated several times in order to achieve the desired mechanical and conductive properties (Figure 1C). After the last PEDOT:PSS/PVA deposition, the sample is baked at 75 °C for 20 min to ensure the complete evaporation of the solvent. All the electrodes characterized in this paper have been fabricated using nine layers and with the same planar size, namely 15 mm × 40 mm.

After the fabrication, in order to further minimize the contact resistance between the ferrimagnetic-conductive part of the film and the external magnetic connection, a thin Ag layer of about 50 nm is thermally evaporated over the portion of the film containing the ferrite particles (Figure 1D). This step allows a great improvement of the electrical connection of the film with the flat, rigid magnetic connector. In fact, as the film is deposited onto an uneven substrate (the human skin), it exposes a limited number of contact points, thus demanding a decrease in the specific contact resistance of the film with the magnet. The film can be eventually peeled-off from the PET carrier and placed on a piece of paper using few droplets of deionized water; from here it can be conveniently transferred onto the skin. The final electrodes are thus characterized by a thin part (the one without the ferrite powder), which offers an optimal interface with the skin, and a thicker, conductive and ferrimagnetic part for the magnetic interconnection, as shown in Figure 1E,F. As reported in the insets in Figure 1E, we performed morphological investigation on the two portions of the film by means of atomic force microscopy (AFM). Both images were performed in semi-contact mode, giving an estimated RMS roughness of 3.8 nm for the ultrathin-conductive part and 120 nm for the ferrimagnetic-conductive part.

## 3. Results and Discussion

### 3.1. Film Characterization

#### 3.1.1. Thickness Characterization

The thickness of both the conductive and the ferrimagnetic-conductive portions of the films was evaluated after each layer deposition using a profilometer (Dektak XT by Bruker, Billerica, MA, USA). For a nine-layer electrode, the thickness of the ferrimagnetic-conductive part was around 30 μm, while the thin conductive part was around 3 μm thick, as shown in Figure 2A, thus ensuring a good skin/electrode interface without the use of any adhesive layer.

#### 3.1.2. Magnetic Force of Interaction

The attractive force of interaction between the film and the magnet has been evaluated by means of a commercial dynamometer (FCA-DS2-50N by IMADA, Northbrook, IL, USA) mounted on a vertical motorized stand (MX2 by IMADA, Northbrook, IL, USA). Figure 2B shows the setup used to measure the attractive force of interaction. A magnet is glued to the end of a flat indenter and placed in contact with the film. The indenter is then slowly moved away from the surface and the attraction force is recorded as the force is measured right before the magnets detaches from the film. In Figure 2C, the quantification of the interaction of the conductive-ferrimagnetic part versus number of layers is presented. For a nine-layer electrode, the maximum measured pressure (a parameter that gives an indication on the magnetic force of interaction) was 12 g/cm^2^, with an average of 9 g/cm^2^ as shown in Figure 2C. This value is sufficient to hold the recording cable of the acquisition instrument in place during static measurements and to ensure a stable recording.

#### 3.1.3. Conductivity Characterization

The conductivity of both the conductive and the ferrimagnetic-conductive part of the structure was evaluated before the silver deposition using a 4284A LCR-meter (by Agilent, Santa Clara, CA, USA). The probes were placed at a lateral distance of 10 mm, one on the top face and one on the bottom face of the film in order to consider the contribution of both the vertical and the planar conductivity. This aspect is very important since, in the proposed approach, the external connector is placed on the opposite side with respect to the skin/electrode interface, thus the vertical conductivity must also be adequate. The average conductivity of the thin conductive part was 8.7 ± 0.2 S/cm, while that of the ferrimagnetic-conductive part was 3.2 ± 0.1 S/cm. As expected, the presence of the non-conductive ferrimagnetic particles slightly reduced the conductivity of the ferrimagnetic part. However, the conductivity of the thin conductive part is in line with that already reported for other PEDOT:PSS-based epidermal films [34].

### 3.2. ECG Measurement Setup and Validation Methods

The suitability of the proposed films for biopotential recording applications were preliminary evaluated on a healthy volunteer using a 32-channel Porti7 electrophysiological recording system (TMSi, EJ Oldenzaal, The Netherlands) at a sampling frequency of 2048 Hz, with an effective bandwidth of 553 Hz. The experimental protocol was conducted following the principles outlined in the Helsinki Declaration of 1975, as revised in 2000. The performance of the ferrimagnetic-conductive epidermal electrodes was compared to that of disposable gelled Ag/AgCl electrodes (BlueSensor N by Ambu, Ballerup, Denmark). The two electrode types were exploited for the recording of an ECG signal, simultaneously and repetitively, thus providing a total of seven 30 s long ECG traces in about one hour. The ferrimagnetic-conductive electrodes were contacted using custom-made magnetic connectors with a snap contact for the connection to the instrument shielded cables, as shown in Figure 3.

To better appreciate the QRS complex in the recordings, lead II was chosen for all ECG signal acquisitions following a Holter configuration. As such, the LL electrode was placed on the left anterior axillary line, the RA electrode was placed slightly under the right manubrium, and the ground electrode was placed near the right hip, as shown in Figure 3. The electrode pairs were arranged on the torso in order to preserve the inter-electrode distance, thus avoiding any mismatch in signal amplitude and morphology.

ECG detection performance was evaluated in terms of signal-to-noise ratio (SNR), entity of baseline wandering (BW) artifact, and cross-spectral coherence (SpCoh) between the new ferrimagnetic-conductive electrode and the commercial Ag/AgCl one. Specifically, the SNR estimation was carried out after removing all low-frequency interferences by digitally filtering each signal by a zero-phase 2nd-order IIR Butterworth high-pass filter with cut-off frequency of 0.67 Hz, as typically recommended [35]. On the high-pass filtered ECG, the SNR was computed as shown in Equation (1).
(1)$$\mathrm{SNR}\;\left[\mathrm{dB}\right]=20\log_{10}\left(A_{\mathrm{pp}}/4\sigma \right),$$
where σ denotes the noise standard deviation estimated on the central isoelectric segment of each T-P interval, whereas A_pp_ identifies the peak-to-peak amplitude of the median beat template, that was extracted after delineating each ECG trace by a state-of-the-art wavelet-based algorithm [36]. BW was quantified as the root mean square (RMS) value of the residual of each ECG signal below 0.67 Hz [35]. The overall analysis is shown in the scheme in Figure 3.

In the frequency domain, the similarity of the ECG recordings acquired with the two electrode types was assessed by the magnitude-squared SpCoh. This figure of merit was estimated for each pair of simultaneous ECG recordings acquired by Ag/AgCl and ferrimagnetic-conductive epidermal electrodes, as [37]:(2)$$\mathrm{SpCoh}=\left|P_{\mathrm{xy}}^{2}\right|/\left(P_{x}P_{y}\right),$$
in which P_x_ and P_y_ represent the power spectral estimates for the signal x and y respectively, whereas P_xy_ is the cross power spectral estimate of x and y. As such, the more the SpCoh approaches 1, the higher the similarity between the signals x and y. In this analysis, the Welch’s overlapped averaged periodogram method [38] with 2 s hamming window and 0.5 s overlap was exploited. Frequency–domain results are reported for all spectral contents below 40 Hz, which is the typical cut-off frequency of the screen/printer filters in ECG devices. Such a value allows achieving better quality and still clinically interpretable ECGs by visual inspection [39] when compared the more conventionally recommended cut-off of 150 Hz [35], at least in a low-risk population.

The skin–electrode impedance of the films was evaluated every 10 min for one hour during the ECG recording session with an electrode impedance meter (EIM105-Prep-Check by General Devices, Ridgefield, NJ, USA, operating frequency: 10 Hz) against the parallel of five Ag/AgCl commercial pre-gelled electrodes using the simple model that was already reported in [40] for different kinds of epidermal electrodes. Figure 4 reports the results of skin–electrode contact impedance, SNR, and BW during the recording session. As can be seen, no trend revealing increasing or decreasing values of SNR and BW could be appreciated over time, even though the skin–electrode contact impedance slightly increases over time. Interestingly, the proposed ferrimagnetic-conductive epidermal electrodes showed a skin–electrode contact impedance comparable to that of other ultra-conformable epidermal electrodes [40,41] which are ultimately suitable for good-quality recordings.

As such, the overall time–domain performance of Ag/AgCl and ferrimagnetic-conductive epidermal electrodes was analyzed and compared graphically in terms of boxplots. As shown in Figure 5A,B, the latter suffer from higher BW artifacts. This analysis is consistent with the results reported in Figure 4, showing higher skin–electrode contact impedance values. Nonetheless, when low-frequency interferences are filtered out, as in typical applications [31], the SNR values obtained by the ferrimagnetic-conductive epidermal electrodes are comparable or even higher than those achieved by the gelled Ag/AgCl ones (Figure 5C). In this regard, some slight amplitude differences can be appreciated in the ECG recordings acquired with the different types of electrodes, which can be ascribed to the different sensing area sizes and the close (but not identical) electrode placements. Remarkably, ferrimagnetic-conductive epidermal electrodes allowed recording very high-quality ECG signals even after being applied to the skin for more than one hour (Figure 5D).

The results of the frequency–domain analysis are reported in Figure 6. ECG signals recorded with the ferrimagnetic-conductive epidermal electrodes and gelled Ag/AgCl electrodes showed high coherence, well represented by the SpCoh (Figure 6A) below 30 Hz. Between 30 and 40 Hz, low SpCoh values were obtained; however, the power contributions in that specific band were found to be negligible (i.e., about 10^−5^), as can be seen in Figure 6B. Remarkably, the main PSD peaks reflecting the physiological contributions of heart rate, T wave, P wave and QRS complex [37,38,39,42] are also well represented in all the different recordings (Figure 6B).

## 4. Conclusions

In this work, a new concept for the fabrication of free-standing tattoo electrodes for biopotential recording is presented. The proposed electrodes, based on ultrathin (3 μm thick in the thinner part) composite films made of a combination of PEDOT:PSS, PVA, and ferrite powder, can be easily connected to a measuring unit using simple magnetic connectors, thus preserving their mechanical stability. This connection approach allows us to overcome the ever-present issue of interfacing fragile, ultrathin electrodes to the recording system without compromising the conformability of the film and is different from standard methods used in epidermal electronics applications. The proposed method is extremely versatile and scalable as it allows modulating the intensity of the mechanical connection with the recording system by tuning the amount of ferrimagnetic powder in the blend. Moreover, it allows adapting the geometrical shape of the electrode according to the needs of the application. The preliminary assessment of the electrodes for ECG signal detection revealed a comparable performance with disposable gelled Ag/AgCl electrodes in terms of SpCoh, SNR, and BW, deserving further investigations with a larger sample size. The achieved results pave the way for the possibility of adapting the proposed technology to the detection of other biopotentials where unobtrusiveness and light weight is relevant, such as some electromyographic or electroencephalographic (low-density, frontal) recordings.

## Figures and Tables

**Figure 1 bioengineering-09-00205-f001:**
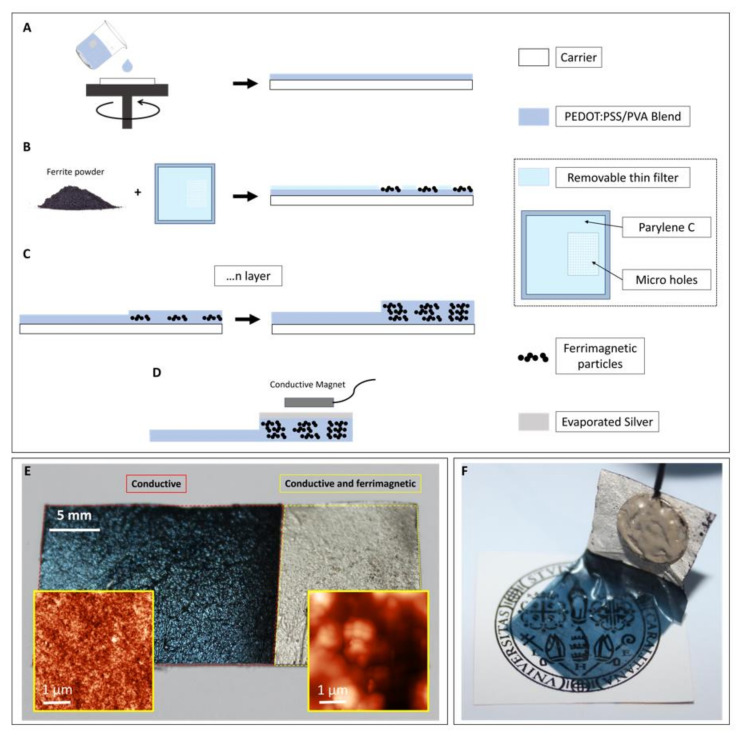
Fabrication process of the ferrimagnetic-conductive film for biopotential recordings. (**A**) A first conductive layer is spin coated onto a PET carrier. (**B**) a first layer of ferrite powder is deposited on the film through a thin Parylene C filter. (**C**) A second conductive layer is spin coated, eventually obtaining a conductive film with a thicker ferrimagnetic-conductive portion. (**D**) A thin Ag layer is evaporated onto the ferrimagnetic-conductive portion of the film in order to improve the magnet–film interface. (**E**) A ferrimagnetic-conductive electrode after the deposition on a piece of standard paper, with the thin, conductive part for an optimal skin/electrode interface and the thicker part, both conductive and ferrimagnetic for the magnetic connection. In the inset: AFM micrographs of both film areas. (**F**) A ferrimagnetic-conductive electrode hanging from a magnet after the peel-off from the PET carrier.

**Figure 2 bioengineering-09-00205-f002:**
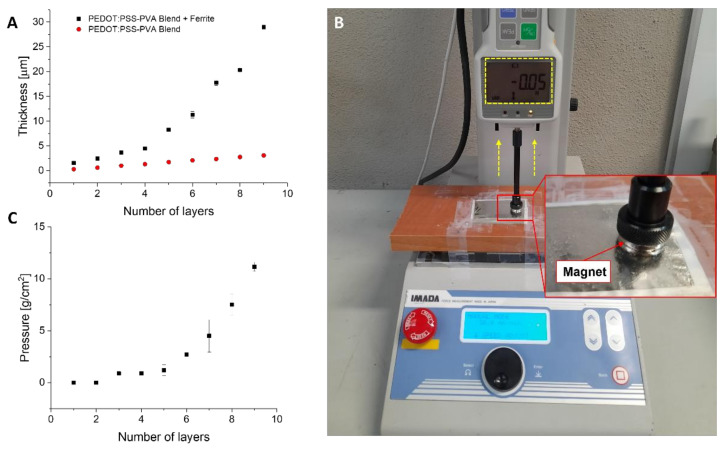
(**A**) Characterization of the thickness of both portions of the film with respect to the number of layers. (**B**) Setup for the measurement of the attractive force of interaction. (**C**) Quantification of the interaction (measured as g/cm^2^) with respect to the number of layers.

**Figure 3 bioengineering-09-00205-f003:**
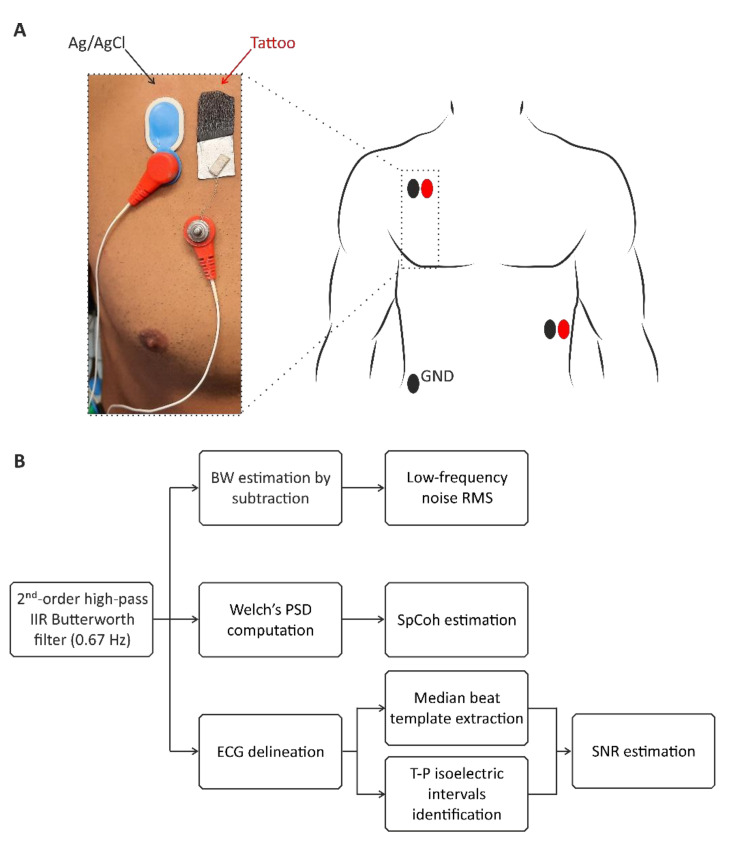
(**A**) The experimental setup used to measure the ECG. The ferrimagnetic-conductive electrodes are contacted using a custom magnetic connector with a snap contact for the connection with the recording instrument. (**B**) Processing chain of the ECG signals. After the acquisition, the signals are firstly high-pass filtered by a digital zero-phase 2nd-order IIR Butterworth filter with cut-off frequency at 0.67 Hz. After that, on the one hand, the SNR is computed by performing ECG delineation to identify a median beat template and the central portion of T-P isoelectric intervals, while on the other hand, the cross spectral coherence is analyzed by considering Welch’s overlapped averaged power spectral densities (PSDs). Finally, BW is quantified by subtracting the high-pass signal from the corresponding raw signal and evaluating the RMS.

**Figure 4 bioengineering-09-00205-f004:**
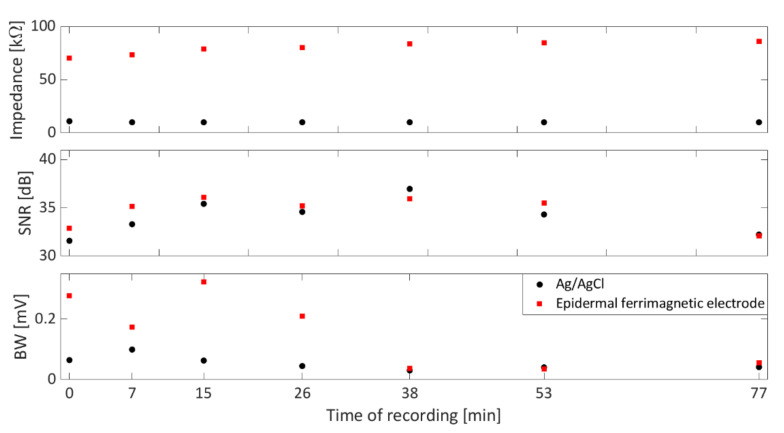
Trends for the skin–electrode contact impedance (**top**), SNR (**middle**), and BW (**bottom**) evaluated at the different recording times on the ECG signals recorded by Ag/AgCl electrodes (●) and the ferrimagnetic-conductive epidermal electrodes (■). The 0 min time of recording refers to the first ECG acquisition, while the last time of recording (77 min) refers to the last recording session, which was performed on the same subject, after 77 min.

**Figure 5 bioengineering-09-00205-f005:**
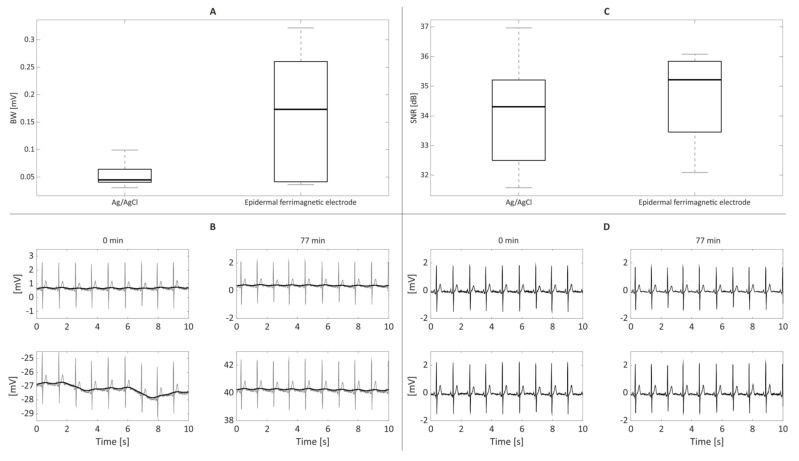
On the left, BW results (**A**) are reported along with the raw ECG signals ((**B**), in grey) recorded at the first (i.e., 0 min, left column) and last recording sessions (i.e., after 77 min, right column) with the corresponding BW artifact (in black) for the Ag/AgCl ((**B**), top) and proposed ferrimagnetic-conductive epidermal electrodes ((**B**), bottom). On the right, SNR results for the different electrode types (**C**) are depicted, while in (**D**) the filtered ECG signals recorded when the recording session started (0 min, left column) and at its end (i.e., after 77 min, right column) by exploiting the Ag/AgCl ((**D**), top) and proposed ferrimagnetic-conductive epidermal electrodes ((**D**), bottom) are presented.

**Figure 6 bioengineering-09-00205-f006:**
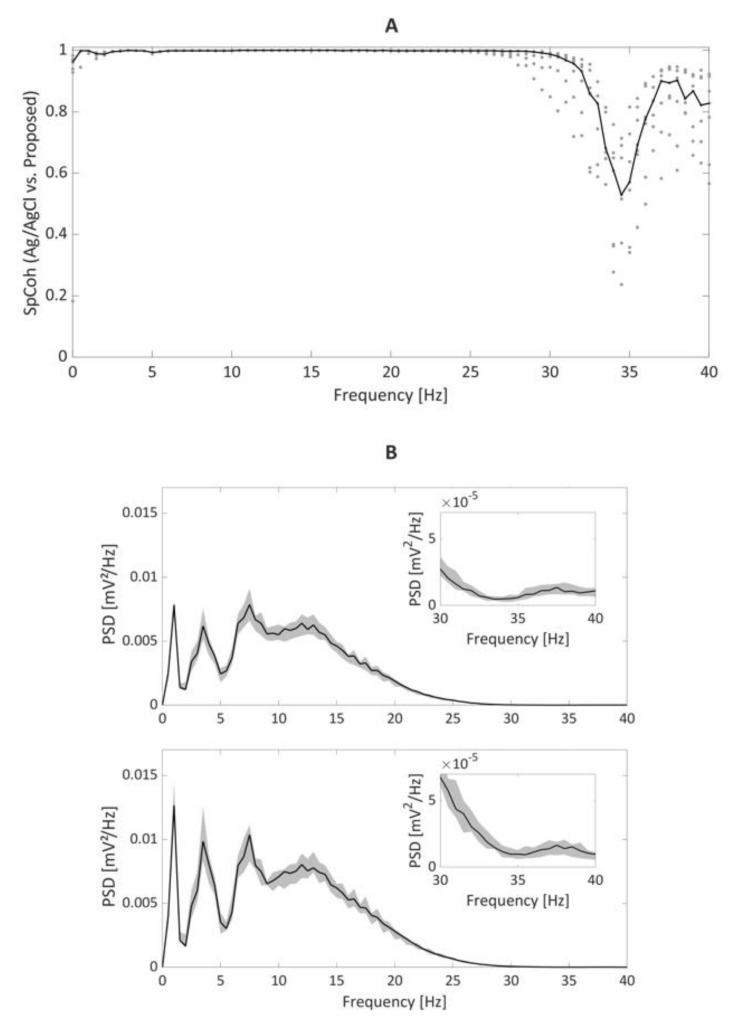
SpCoh results (**A**) for the comparison of the recording obtained by gelled Ag/AgCl and ferrimagnetic-conductive epidermal electrodes recordings ((**A**), on the left). Black line represents the median values of SpCoh across frequencies, whereas grey dots represent the values assumed in the different recording sessions. In (**B**), the median PSD trends (in black) with their 5th and 95th percentiles (grey shading) are depicted for the recordings with Ag/AgCl (top) and the proposed ferrimagnetic-conductive epidermal electrodes (bottom), along with an enlarged depiction of the 30–40 Hz range.

## Data Availability

The data supporting the conclusion of this article are available from the authors upon request, without undue reservation.

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
