# Peer review of "Epidermal Electrodes with Ferrimagnetic/Conductive Properties for Biopotential Recordings"

_bioengineering, 2022, doi:10.3390/bioengineering9050205_

Round 1

Reviewer 1 Report

This paper reports on application of ferromagnetic particles for fabrication of epidermal electrodes to improve their connection. This investigation is interesting and could be considered for publication on Bioengineering.

Author Response

We thank the reviewer for the positive comments on our work. We hope that the changes we made in this revised version further improved the quality of the manuscript.

Reviewer 2 Report

In this work, the authors prepared the free-standing tattoo electrodes for biopotentials recording. The electrodes were composed of the PEDOT:PSS, PVA and ferrite powder, which can ensure a facile connection with the external magnetic connectors for optimization of the mechanical stability. The assessment of the electrodes for ECG signals also revealed the potential for applications in real-life. My comments in details are as below.

  1. In line 112, page 3, what is the ‘highest PVA content’? Please clarify.
  2. In Figure 1C, how to prevent the flow of the un-cured film as it is uneven, which can be clearly observed from the schematic diagram? Furthermore, it is not clear why such a film thickness (Figure 2A) was selected as the standard? Is there any control experiment to verify the advantage of such thickness design?
  3. In Figure 1E, the scale bars should be provided to reveal the dimension of the sample.
  4. It is unclear why a silver layer can reduce the overall resistance when considering the magnet, the silver layer, and the ferrimagnetic part together. They are in the series connection, so the total resistance should be affected by the one with relatively larger resistance. Please provide direct comparison between the one without and with the silver layer.
  5. For the filter, why the hole diameter of 50 µm was used? If the hole diameter was tuned, what will be the overall performance of the as-prepared composite?
  6. In Figure 2B, I am wondering why a unit of g/cm[2], instead of N/cm[2], was used for the pressure.
  7. There are no SEM images for the ferrimagnetic particles, the as-prepared film surface, and so on. Furthermore, apart from the film thickness, a surface roughness mapping (e.g. via AFM) should be provided to give the overall evenness of the film. As observed from Figure 1E, the film surface is not uniform based on the methodology described.
  8. The authors indicated that the method could help to improve the stability of the magnetic connection. However, there are no direct proof or comparison that can reveal the significance of the current work. For example, what will be the potential applications of the current work that requires a high mechanical stability? Please clarify.

Author Response

Reviewer 2

In this work, the authors prepared the free-standing tattoo electrodes for biopotentials recording. The electrodes were composed of the PEDOT:PSS, PVA and ferrite powder, which can ensure a facile connection with the external magnetic connectors for optimization of the mechanical stability. The assessment of the electrodes for ECG signals also revealed the potential for applications in real-life. My comments in details are as below.

Rev 2 - Question 1

In line 112, page 3, what is the ‘highest PVA content’? Please clarify.

Rev 2 - Answer 1

We thank the reviewer for spotting this imprecision. The “highest PVA content” refers to the ink formulation that we used for the first layer of the structure. Since the PVA helps the final peel-off of the film but, at the same time, it increases the electrical resistance of the film itself, we decided to employ a slightly higher concentration of PVA only at the beginning of the process, while reducing it in the following steps.

To better point out this aspect, we slightly modified the sentence: 

OLD SENTENCE: In the first step of the fabrication process, the PEDOT:PSS/PVA blend with the highest PVA content is deposited through spin coating (spin speed = 700 rpm, spin time = 60 seconds) on a PET substrate, which acts as a carrier (Figure 1A).

NEW SENTENCE: In the first step of the fabrication process, the PEDOT:PSS/PVA blend with 7% of PVA is deposited through spin coating (spin speed = 700 rpm, spin time = 60 seconds) on a PET substrate, which acts as a carrier (Figure 1A).

Rev 2 - Question 2

In Figure 1C, how to prevent the flow of the un-cured film as it is uneven, which can be clearly observed from the schematic diagram? Furthermore, it is not clear why such a film thickness (Figure 2A) was selected as the standard? Is there any control experiment to verify the advantage of such thickness design?

Rev 2 - Answer 2

This remark allows us to better explain this part, which might be misleading. In fact, no actual flow of the ink occurs during this part of the fabrication. The ink is spin coated on the surface and during the 60 s of the process most of the solvent evaporates, thus leaving an almost formed thin film. The curing process is only needed to let the solvent completely evaporate. Moreover, the thickness difference between the two parts of the film is due to the addition of the ferrite powder that follows each spin coating step. The ferrite is localized on the desired portion of the surface thanks to the use of a patterned physical filter (a microperforated ultrathin Parylene C membrane).

As regards the thickness, we evaluated it by focusing on the desired attraction force, which was based on the weight of the cables used for the validation, and we decided to follow this path because our goal was mainly to present the concept. Different thicknesses could be of course obtained using different ferrimagnetic/ferromagnetic/magnetic particles for example, and the minimum required force of attraction might change as a function of the desired application. This aspect highlights, in our opinion, the flexibility and versatility of the proposed approach.

Rev 2 - Question 3

In Figure 1E, the scale bars should be provided to reveal the dimension of the sample.

Rev 2 - Answer 3

We again thank the reviewer for spotting this error. We modified the figure according to his/her suggestion.

Rev 2 - Question 4

It is unclear why a silver layer can reduce the overall resistance when considering the magnet, the silver layer, and the ferrimagnetic part together. They are in the series connection, so the total resistance should be affected by the one with relatively larger resistance. Please provide direct comparison between the one without and with the silver layer.

Rev 2 - Answer 4

We’d like to thank the reviewer to rise this question, since he/her allows us to better explain this very important aspect of our work. Indeed, the mentioned resistance contributions are in series thus their resistances sum up and the larger of the three may affect the final value. Exactly for this reason, the presence of the silver layer has a very significant role. In fact, without it, the ferrimagnetic layer resistance may be considered as the sum of the ferrimagnetic bulk plus the contribution of the contact resistance at the interface between the magnet and the surface of the ferrimagnetic layer. As the last is very rough, this prevents a real conformal contact with the magnet surface (that is flat and rigid) and this has the consequence that the conductive paths from the patch to the magnetic connector are limited by a relatively few contact points formed where the ferrimagnetic layer physically touches the rigid magnet surface. Moreover, these contact points have a high concentration of ferrimagnetic particles and thus are characterized by a high electrical resistance. Adding a metal layer increases the area and the conductance of the material in correspondence of the contact points and in addition allows the collection of charges from the areas of the film surrounding the contact points. Therefore, the global effect is to decrease the (dominant) overall contact resistance because of the improvement of the conductance of the contact layer between the patch and the magnet.

To back our speculations, we performed resistance measurements on bare ferrimagnetic-conductive films and Ag-covered ferrimagnetic conductive films, with the setup showed in the figure (panel A). The results (shown in panel B) demonstrate a considerable resistance reduction (an order of magnitude) for the Ag-covered films, as expected. Please refer to the attched PDF for the figure.

Rev 2 - Question 5

For the filter, why the hole diameter of 50 µm was used? If the hole diameter was tuned, what will be the overall performance of the as-prepared composite?

Rev 2 - Answer 5

We agree with the reviewer that, as the main role of the filter is to create an even distribution of the particles in the film without using any chemical modification of the particles themselves, the diameter of the holes (as well as their density and, even more importantly, the thickness of the filter itself) might have an impact on the overall performance of the composite. However, a thorough analysis of the effect of the dimension of the holes was beyond the scope of this article. To put our choice in perspective, we empirically selected the filter’s dimensional parameters to avoid the formation of the large lumps that result from the use of the as-prepared composite (i.e. obtained by directly mixing the particles with the ink).

Rev 2 - Question 6

In Figure 2B, I am wondering why a unit of g/cm[2], instead of N/cm[2], was used for the pressure.

Rev 2 - Answer 6

We thank the reviewer for the remark. We used this particular unit to better highlight the actual weight that the film can hold using a fixed magnet area. Similar units (kPa) are commonly used in this kind of characterization (Jang , K. et al. Ferromagnetic, Folded Electrode Composite as a Soft Interface to the Skin for Long-Term Electrophysilogical Recording. 2016, Adv. Funct. Mater., 26, 7281–7290, DOI: 10.1002/adfm.201603146).

Rev 2 - Question 7

There are no SEM images for the ferrimagnetic particles, the as-prepared film surface, and so on. Furthermore, apart from the film thickness, a surface roughness mapping (e.g. via AFM) should be provided to give the overall evenness of the film. As observed from Figure 1E, the film surface is not uniform based on the methodology described.

Rev 2 - Answer 7

We thank the reviewer for the suggestion, which gives us the chance of improving the quality of the manuscript. We performed an AFM mapping of both the ultrathin-conductive and the ferrimagnetic-conductive portions of the film and modified Figure 1 accordingly. We also slightly modified the manuscript adding  the estimated values of the RMS roughness in both cases (i.e. for the ultrathin-conductive part and the ferrimagnetic-conductive part of the film):

“As reported in the insets in Figure 1E, we have performed morphological investigation on the two portions of the film by means of atomic force microscopy (AFM). Both images have been performed in semi-contact mode giving an estimated RMS roughness of 3.8 nm for the ultrathin-conductive part and 120 nm for the ferrimagnetic-conductive part.”

Rev 2 - Question 8

The authors indicated that the method could help to improve the stability of the magnetic connection. However, there are no direct proof or comparison that can reveal the significance of the current work. For example, what will be the potential applications of the current work that requires a high mechanical stability? Please clarify.

Rev 2 - Answer 8

The scientific problem that this work tries to solve is, in itself and in our opinion, of great relevance. In fact, as far as we know, this approach provide an unprecedented solution for a practical interfacing of ultrathin epidermal electrodes, a problem that constitutes probably the main issues when it comes to tattoo electronics. A possible follow up of this work can be the realization of a lightweight, wearable, comfortable and wireless electronics that can be magnetically connected to these epidermal electrodes for the detection of ECG, EEG and EMG in dynamic conditions, a field that is considerably limited by the presence of cumbersome connectors that, ultimately, balance out the potential benefits of having an ultrathin, imperceptible recording device.

Reviewer 3 Report

Dear Editor,

There are almost sufficient works have been done in this paper with the title of "Epidermal electrodes with ferrimagnetic/conductive properties for biopotential recordings" to achieve a useful product which in my view it is good to have accepted in the journal of Bioengineering. However, there are some parts that needed to be corrected which are sorted below:

  1. The abstract is incomplete, it should contain all of the detailed information about the prominent achievements of the experiments including numeric results.
  2. In the introduction, goals should be mentioned in the latest paragraph but the authors report on the work process. It needs to be changed.
  3. In the experimental section, Specifications of some devices are not included. Please correct.
  4. In the experimental section, it is better to depict any test with a flowchart or simple schematics.
  5. In the introduction section, it should be better to talk about the importance of past work. My suggestion is to use this reference:

Najafi et al, Synthesis and Magnetic Properties Evaluation of Monosized FeCo Alloy Nanoparticles Through  Microemulsion Method, Journal of Superconductivity and Novel Magnetism, vol.30, number9, 2017

  1. In the conclusion, the writer should bring their numeric results with their discussions altogether.
  2. Unfortunately, I could not find any novelty in this work. Please elucidate it.

Author Response

Reviewer 3

Dear Editor,

There are almost sufficient works have been done in this paper with the title of "Epidermal electrodes with ferrimagnetic/conductive properties for biopotential recordings" to achieve a useful product which in my view it is good to have accepted in the journal of Bioengineering. However, there are some parts that needed to be corrected which are sorted below:

Rev 3 - Question 1

  1. The abstract is incomplete, it should contain all of the detailed information about the prominent achievements of the experiments including numeric results.

Rev 3 – Answer 1

We thank the reviewer for the suggestion. We slightly modified the abstract by adding the thickness value and the figure of merits used for the comparison of the tattoo performance with that of commercial Ag/AgCl.

NEW ABSTRACT: Interfacing ultrathin functional films for epidermal applications with external recording instruments or readout electronics still represents one of the biggest challenges in the field of tattoo electronics. With the aim of providing a convenient solution to this ever-present limitation, in this work we propose an innovative free-standing electrode made of a composite thin film based on the combination of the conductive polymer PEDOT:PSS and ferrimagnetic powder. The proposed epidermal electrode can be directly transferred on the skin and is structured in two parts, namely a conformal conductive part with a thickness of 3 μm and a ferrimagnetic-conductive part that can be conveniently contacted using magnetic connections. The films were characterized for ECG recordings, revealing a performance comparable to that of commercial pre-gelled electrodes in terms of cross-spectral coherence, signal-to-noise-ratio, and baseline wandering. These new conductive, magnetically-interfaceable and free-standing conformal films introduce a novel concept in the domain of tattoo electronics and can set the basis for the development of a future family of epidermal devices and electrodes.

Rev 3 - Question 2

  1. In the introduction, goals should be mentioned in the latest paragraph but the authors report on the work process. It needs to be changed.

Rev 3 - Answer 2

We agree with the reviewer that the goals should be better highlighted in the introduction. We added a sentence at the beginning of the last paragraph, according to the suggestion:

“With the intent of overcoming the aforementioned issues and thus providing an innovative approach for epidermal electrodes interfacing, we developed a simple fabrication process with which is possible to obtain ultrathin functional films that can be easily contacted using magnetic connectors.”

However, we stand by the choice of reporting on the reasons behind the selection of the materials in the same paragraph, as we think that the readers would need some context and references to better get the following description of the employed solution.

Rev 3 - Question 3

  1. In the experimental section, Specifications of some devices are not included. Please correct.

Rev 3 - Answer 3

We thank the reviewer for spotting this problem, we fix it by adding the specification of the impedance measurements (the operating frequency was missing).

Rev 3 - Question 4

  1. In the experimental section, it is better to depict any test with a flowchart or simple schematics.

Rev 3 – Answer 4

We thank the reviewer for the suggestion, which gives us the opportunity to better depict the signal analysis part and significantly improve the readability of the paper. We modified Figure 3 adding a flow chart of the overall analysis performed on the recorded ECG signals.

Rev 3 - Question 5

  1. In the introduction section, it should be better to talk about the importance of past work. My suggestion is to use this reference:

Najafi et al, Synthesis and Magnetic Properties Evaluation of Monosized FeCo Alloy Nanoparticles Through  Microemulsion Method, Journal of Superconductivity and Novel Magnetism, vol.30, number9, 2017

Rev 3 - Answer 5

We thank the reviewer for the suggestion. We slightly modify the introduction accordingly and added the suggested reference, which helps us to better contextualize the choice of the employed fabrication approach:

“Moreover, the simple approach presented in this work allows to obtain high-performing films without any chemical modification of the particles (such as for example that reported by Najafi et al. [29]).”

Rev 3 - Question 6

  1. In the conclusion, the writer should bring their numeric results with their discussions altogether.

Rev 3 - Answer 6

We slightly modify the conclusions by adding the thickness and the actual figure of merit that we employed in the comparison with the commercial electrodes.

Rev 3 - Question 7

  1. Unfortunately, I could not find any novelty in this work. Please elucidate it.

Rev 3 – Answer 7

The novelty of the work (which is now hopefully more clear after the slight modification of the introduction) resides in the completely new approach to a long-standing problem in the field of epidermal electronics, namely how to conveniently interface ultrathin films once they are positioned on the skin. In fact, the rapid advancement of the field in terms of new materials and functionalities has not been supported by an equally rapid advancement of the connection techniques, this aspect being the main limiting factor that prevented these new technology from being more widespread. The concept behind the proposed approach offers an unprecedented “out of the box”-like solution to the problem, which will hopefully encourage researchers in our community to explore and expand this idea. That’s why we think that our work presents a high level of innovation.

Round 2

Reviewer 2 Report

The authors have addressed my concerns with solid experimental supports. I recommend the acceptance of this work in Bioengineering.

Reviewer 3 Report

Dear Editor,

Corrections have been made and the article has been accepted.